# The Usability of Pumice Powder as a Binding Additive in the Aspect of Selected Mechanical Parameters for Concrete Road Pavement

**DOI:** 10.3390/ma12172743

**Published:** 2019-08-27

**Authors:** Abdulrezzak Bakis

**Affiliations:** Department of Civil Engineering, Bitlis Eren University, Bitlis 13100, Turkey; arezzakbakis@gmail.com

**Keywords:** pumice, lime, concrete road pavement, compressive strength, bending strength

## Abstract

In this study, the usability of pumice powder and lime in concrete production as a binding additive for rigid superstructure concrete road pavement was investigated. Following the determination of the optimum binder ratio, these new binder ratios were used in crushed limestone concrete production. The concrete thus formed was named concrete containing cement, pumice powder and lime (PPCC). The normally produced concrete, without pumice powder and lime binder was selected as reference concrete (RC). Regarding the total binder amount of the most appropriate binder ratio 50% was found to be cement, 30% pumice powder and 20% lime in the result of the study. In consequence of the study, the 20 ± 2 °C and 7–28 days compressive strengths of the reference concrete were found to be 33.8 MPa and 38.2 MPa and its bending strengths were 4.2 MPa and 4.7 MPa. The 20 ± 2 °C and 7–28 days compressive strengths of PPCC were found to be 25.1 MPa and 28.3 MPa and its bending strengths were 3.2 MPa and 3.5 MPa. The results of the study showed the usability of PPCC in concrete pavement.

## 1. Introduction

The term “pumice” is called “ponce” in French, whereas the stones with medium particle size are called “pumice” in English [1]. Pumice stone formations, which are formed because of volcanic events and have a cavernous, spongy structure, are found in many regions of the world where volcanic activities take place [2]. Pumice contains numerous pores ranging from macro scale to micro scale due to the sudden release and sudden cooling of the gases it embodies during its formation. Since there are disconnected caverns between the pores, its permeability is low and heat and sound insulation is quite high [3]. Today, the use of pumice is developing day by day when compared with the past, and it is being used in various fields. Its usage in other sectors is newly becoming widespread [4]. Pumice sources identified around the world are approximately 18 billion m^3^ [5]. Especially regarding pumice beds, Bitlis province has significant potential due to both volcanic area and geological structure. The beds in question are located in the Tatvan district of Bitlis province and 81,500,000 m^3^ pumice beds of good quality are available [6,7]. As the pumice grain grows, the grain specific gravity decreases. Pore percentage increases as grain sizes increase. Pumice is a very light, pyroclastic magmatic rock type shaped during the volcanic eruption. The lava is shaped in liquid form, including gas bubbles, throughout the period in which it spurts out into the air as gas froth [8]. Pumice is especially used in the production of trass cement. When pumice stones are ground with cement fineness and then mixed with cement or lime, they acquire a binding property. These types of volcanic rocks are called pozzolana [9]. Small crystals of various minerals are found in pumices, which have an amorphous structure. The most common crystals are feldspar, augite, hornblende and zircon. Pumice is much lighter than sand and gravel [10]. Pumice sand is one of the solid wastes polluting the environment and causing the formation of dumpsites.

When processing pumice extracted from quarries, a significant amount of pumice sand is produced. Waste materials should be evaluated for their possible uses. Pumice is a pyroclastic rock type [11]. Pumice contains several independent pores, each separated from the other by a glassy membrane [12]. The construction of rigid pavements has been rapidly becoming widespread in airports, in structures like runways, in terminals, loading, unloading and parking areas and partially in places like urban roads, etc.

Considering the increasing number of commercial vehicles in the world with each passing day, the use of concrete pavements as road superstructures is expected to become widespread in the future [13]. Overall, it is almost imperative that roads, bearing 8.2 t of standard axle load number of which are more than 60–75 × 10^6^ within a 20-year period and airports with more than 5000 departures of large passenger airplanes yearly, are constructed as rigid pavements. In the specifications for the proportioning of concrete pavement, a few of the criteria considered are maximum w/c ratio being 0.40–0.45, minimum compressive strength being 280 kg/cm^2^ and minimum amount of cement being 270–335 kg/m^3^ [14]. In buildings and similar structures, water/cement ratio is expected not to exceed 55% [15]. Materials that are initially thin, which turn into dough when water is added, and that do not dissolve in water by hardening over time are called binding materials.

The use of these materials dates back to ancient times. The most commonly used binders are oily lime, cement and gypsum. Before cement was found in the 19th century, soil and limestone mixtures, different limes and gypsums, baked clay and other binders (volcanic tuffs, trass, zeolite, diatomite and natural soils) were used as binding materials [16]. The materials that can harden under water are called hydraulic binders. The use of these binders coincides with the Hellenistic period of 2300 years ago. In the town of Pozzuoli, near Naples, it was discovered by the Romans that this soil, made up of ashes emanating from the Vesuvius volcano, was superior to plain lime mortar when mixed with lime [17]. Natural or artificial additives such as pozzolanic substance, which do not exhibit binding properties on their own but acquire binding property in aqueous media when mixed with lime in a very finely ground state are called pozzolanic additives.

Since the compressive strength of pumice, which has a significant amount of pores in macro and micro dimensions is very low, its areas of usage in the construction sector have remained limited and it has mainly been used in wall and plaster construction aimed at insulation, in filter material production in water treatment units, in grinding and polishing jobs in the textile industry, in torching material production in match factories and in similar sectors. Pumice aggregate is finitely used for insulation purposes in concrete production sector, but it cannot be used as building concrete in the carrier parts of the structure since the compressive strength of the aggregate is very low. This study was prepared with the aim of providing the widespread use of concrete elements that will be formed with pumice powder in the construction sector. It is possible to use other materials with a binding property such as pozzolanic rock powders in certain ratios together with cement in order to reduce the cost in concrete production [13]. In their study, Hattatoglu and Bakis (2017) used ignimbrite powder as a binding additive and thus obtained high-strength concrete [18].

In this study, pumice powder and lime were found to have a significant binding effect on concrete when mixed with cement. In this study, a new type of concrete production was developed by using pumice powder (PP) and lime (L) together with cement. The compressive and bending strengths of the concrete samples produced in this way were found to be high and the concrete production cost was reduced by reducing the amount of cement per m^3^ in production. In this study, cement, crushed stone (gravel-sand) and water containing normal concrete production, which are generally used in construction sector, were prepared for the purpose of comparison and the samples to be formed were selected as reference concrete.

In this study, the usability of pumice powder and lime in concrete production as a binding additive for concrete road pavement was investigated. Pumice powder with dimensions of 0–0.04 mm was used in the study. Seventy-two types of concrete samples were formed in different mixing ratios, with pumice powder. The optimum ratios of pumice powder and lime as the binding additive were determined in consequence of all the experiments. Following the determination of the optimum binder ratio, these new binder ratios were used in crushed limestone concrete production. Compressive and bending strength tests of the new concrete produced were performed. The concrete thus formed was named concrete containing cement, pumice powder and lime (PPCC). The concrete produced without pumice powder and lime binder, only with cement binder was selected as reference concrete (RC). RC and PPCC concrete were cured with standard water curing of 7 and 28 days. Following water curing, compressive and bending strength tests were performed on all concrete samples. The results of the study showed the usability of PPCC in concrete pavement.

## 2. Materials and Methods

### 2.1. Materials

CEM I 42.5 R type cement, which complies with TS EN 197-1 (EN 197-1:2011) (2012) standard, was used in all experiments [19]. Chemical properties of CEM I 42.5 R cement are given in Table 1 [20].

The cement appearance is given in Figure 1.

Potable Bitlis city water was used in the experiments. Pumice powder is shown in Figure 2.

Pumice powder grain diameter was between 0–0.04 mm. Specific gravity of slacked lime was 2.2 g/cm^3^ (Miner Mining Transportation Trade Limited Company) [21] and complied with the TS EN 459-1 (2017) (EN 459-1:2015) standard [22]. The lime view is shown in Figure 3.

The chemical properties of pumice and lime are shown in Table 2 [23,24].

The pumice powder used as cement additive must comply with TS 25 (TS 25/T1) (2008) Trass Standard. It was indicated in the TS 25 (TS 25/T1) (2008) Trass Standard prepared by TSE (Turkish Standardization Institute) that the SiO_2_ + Al_2_O_3_ + Fe_2_O_3_ total should at least have the ratio of 70% [25]. As shown in Table 2, the pumice powder SiO_2_ + Al_2_O_3_ + Fe_2_O_3_ was 86.5% in total. This ratio indicates that pumice powder can be used as binder. For comparison purposes, physical and mechanical properties of pumice, cement, and lime are shown in Table 3 [26,27].

One of the highest cost items in concrete production is the amount of cement used in production. In this study, the compressive strengths of samples formed by using pumice powder (PP) and lime (L) together with cement (C) were calculated in order to reduce the amount of cement. Seventy-two different mixture types were formed for the determination of the optimum binder ratio. With all the mixtures formed to determine the appropriate binder ratio, the water/binder ratio was considered as 0.60. Consistency and workability were not influenced by the substitution. Three samples from each type of mixture were taken, and the average of these three values was calculated. The prepared mixtures are shown in Figure 4.

### 2.2. Methods

#### 2.2.1. Binder Mixing Ratios

##### Type-1 Mixing Ratios

Type-1 mixing ratios are shown in Table 4. The mixture contained cement, pumice powder and water. There was no lime in the mixture.

Six different types of mixture were formed by taking 0%, 20%, 40%, 60%, 80% and 100% of the cement amount.

As shown in Table 4 and Figure 5, the 1-1 mixture was the binder’s reference mortar for optimal binder fixation. Only cement was used as binder in the reference mortar.

As shown in Table 4 and Figure 6, only pumice powder was used as the binder in the mixture mortar 1-6.

##### Type-2 Mixing Ratios

Type-2 mixing ratios are shown in Table 5. The mixture contained pumice powder, lime and water. There was no cement in the mixture.

Six different types of mixture were formed by taking 0%, 20%, 40%, 60%, 80% and 100% of the pumice powder amount as lime amount.

As shown in Table 5 and Figure 7, only lime was used as binder in the mixture mortar 2-6.

##### Type-3 Mixing Ratios

Type-3 mixing ratios are shown in Table 6. The mixture contained cement, lime and water. There was no pumice powder in the mixture.

Six different types of mixture were formed by taking 0%, 20%, 40%, 60%, 80% and 100% of the cement amount as lime amount.

##### Type-4 Mixing Ratios

Type-4 mixtures included cement, pumice powder, lime and water. The mixing ratio of each type was different. Nine different types of mixtures were produced from Type-4 mixtures.

##### Type-4-1 Mixing Ratios

Type 4-1 mixing ratios are shown in Table 7. In this section, the mixing ratio with the highest compressive strength was considered.

In the mixture ratio with the highest compressive strength, six different types of mixture were formed by considering the amount of lime as 0%, 20%, 40%, 60%, 80% and 100% of the cement amount.

##### Type-4-2 Mixing Ratios

Type-4-2 mixing ratios are shown in Table 8. In this section, the mixing ratio with the highest compressive strength in the second type mixture was considered.

In the mixture ratio with the highest compressive strength, six different types of mixture were formed by considering the amount of lime as 0%, 20%, 40%, 60%, 80% and 100% of the pumice powder amount.

##### Type-4-3 Mixing Ratios

Type-4-3 mixing ratios are shown in Table 9. In this section, the mixing ratio with the highest compressive strength in the first type mixture was considered.

In the mixture ratio with the highest compressive strength, six different types of mixture were formed by considering the amount of lime as 0%, 20%, 40%, 60%, 80% and 100% of the (cement + pumice powder) amount.

##### Type-4-4 Mixing Ratios

Type-4-4 mixing ratios are shown in Table 10. In this section, the mixing ratio with the highest compressive strength in the second type mixture was considered.

In the mixture ratio with the highest compressive strength, six different types of mixture were formed by considering the amount of cement as 0%, 20%, 40%, 60%, 80% and 100% of the pumice powder amount.

##### Type-4 Mixing Ratios

Type-4-5 mixing ratios are given in Table 11. In this section, the mixing ratio with the highest compressive strength in the second type mixture was considered.

In the mixture ratio with the highest compressive strength, six different types of mixture were formed by considering the amount of cement as 0%, 20%, 40%, 60%, 80% and 100% of the lime amount.

##### Type-4-6 Mixing Ratios

Type-4-6 mixing ratios are presented in Table 12. In this section, the mixing ratio with the highest compressive strength in the second type mixture was considered.

In the mixture ratio with the highest compressive strength, six different types of mixture were formed by considering the amount of cement as 0%, 20%, 40%, 60%, 80% and 100% of the (pumice powder + lime) amount.

##### Type-4-7 Mixing Ratios

Type-4-7 mixing ratios are given in Table 13. In this section, the mixing ratio with the highest compressive strength in the third type mixture was considered.

In the mixture ratio with the highest compressive strength, six different types of mixture were formed by considering the amount of pumice powder as 0%, 20%, 40%, 60%, 80% and 100% of the cement amount.

##### Type-4-8 Mixing Ratios

Type-4-8 mixing ratios are shown in Table 14. In this section, the mixing ratio with the highest compressive strength in the third type mixture was considered.

In the mixture ratio with the highest compressive strength, six different types of mixture were formed by considering the amount of pumice powder as 0%, 20%, 40%, 60%, 80% and 100% of the lime amount.

##### Type-4-9 Mixing Ratios

Type-4-9 mixing ratios are shown in Table 15. In this section, the mixing ratio with the highest compressive strength in the third type mixture was considered.

In the mixture ratio with the highest compressive strength, six different types of mixture were formed by considering the amount of pumice powder as 0%, 20%, 40%, 60%, 80% and 100% of the (cement + lime) amount.

Samples prepared for optimum binder fixation were prepared for 7-day compressive strength determination and each mixture was prepared as three pieces in total, with dimensions of 150 × 150 × 150 mm. Samples were cured for 7 days at 20 ± 2 °C by standard water curing. Samples taken into the curing pool are shown in Figure 8.

#### 2.2.2. Reference Concrete Mixing Ratios

In reference concrete production, CEM I 42.5 R type cement, which complied with TS EN 197-1 (EN 197-1:2011) standards, crushed limestone as aggregate and Bitlis city water qualifying as drinking water for concrete mixing water were used. Reference concrete class was taken as C30/37. In the study, reference concrete samples were prepared for 7 and 28 days of daily compressive strength determination, in dimensions of 150 × 150 × 150 mm. being three each and six in total. Three samples were cured with a 20 ± 2 °C standard water curing of 7 days, and the other three samples were cured with a 20 ± 2 °C standard water curing of 28 days. The curing pool is shown in Figure 8. The quantities of reference concrete materials are shown in Table 16.

Six pieces of samples with dimensions of 100 × 100 × 400 mm were prepared for bending strength. three samples were treated with a 20 ± 2 °C standard water curing of 7 days, and the other three samples were treated with a 20 ± 2 °C standard water curing of 28 days. The curing pool is shown in Figure 8. Compressive and bending strength tests of samples after curing were performed. TS EN 12390-3 (2010) standard (EN 12390-3/2001) [28] was used in the compressive strength test, and the TS EN 12390-5 (2010) standard (EN 12390-5:2000) [29] was used in the bending strength test.

#### 2.2.3. Mixing Ratios of Concrete with Optimum Binding Ratio (PPCC)

In the production of PPCC, CEM I 42.5 R type cement, 0–0.04 mm pumice powder and lime were used as binders in accordance with TS EN 197-1 standards (EN 197-1:2011). Potable Bitlis city water was used as aggregate for crushed limestone and concrete mixed water. C30/37 concrete class was considered in PPCC concrete production. Three samples from each type of mixture were taken, and the average of these three values was calculated. The material mixing ratios obtained at the optimum binder ratio are shown in Table 17.

The water/binder ratio of the concrete (PPCC) prepared in the ratio of reference concrete (RC) and optimum binder were taken as 0.42 as shown in Table 16 and Table 17.

#### 2.2.4. Sieve Analysis Method

The recommended slump value for pavement concrete is 3 cm according to Table 18 [14].

For 3 cm slump, the Water/Cement (W/C) value of which is 0.42, the cement amount was considered as 450 kg/m^3^ and the approximate water amount according to the w/c ratio as 189 kg/m^3^. The mix design target strength for 0.42 w/c ratio was found to be 380 kg/cm^2^ (38 MPa) [14]. Table 19 shows the approximate w/c ratios according to the concrete compressive strengths. Since the maximum w/c ratio in the coating concrete is desired to be between 0.40–0.45, for the non-air entrained concrete, the w/c ratio of which was 0.42, an average target compressive strength was found, the 28 days compressive strength of which was 40 MPa [16,30].

In accordance with Table 19, reference concrete (RC) and optimum binder concrete (PPCC) class were established as C30/37, considering the mean target compressive strength. C30/37 concrete properties can be seen in Table 20 [16,30].

The amount of aggregate required for sieve analysis is given in Table 21 [16,31].

Reference concrete (RC) and optimum binder concrete (PPCC) sieve analysis were performed according to the TS EN 933-1 (2012) (EN 933-1:2012) standard [31]. As stated in TS EN 933-1 (2012) (EN 933-1:2012) for sieve analysis, 3 kg sample was taken considering the largest aggregate grain size in the concrete, which was 16 mm [31].

## 3. Results and Discussion

### 3.1. Compressive Strength Test Results for Optimum Binding Ratio Determination

Samples prepared for compressive strength tests are shown in Figure 9.

The appearance of the samples in the compressive strength tester is shown in Figure 10.

The results are presented for the 7-day period Section 3.1.1, Section 3.1.2, Section 3.1.3 and Section 3.1.4. Three samples for each mixture design were taken, and the average of these three values was calculated.

#### 3.1.1. Type-1 Compressive Strength Test Results

Type-1 mixture quantity, unit volume weight (BHA) and compressive strength test results are shown in Table 22.

Pumice powder was taken as 0%, 20%, 40%, 60%, 80% and 100% of the cement amount. The mixture 1-1 was the reference mortar of the binder paste. Only cement was used as binder in the reference mortar. According to the reference mortar, the mixture with the highest strength was the 1-2 mixture and its compressive strength was found to be 17.3 MPa.

#### 3.1.2. Type-2 Compressive Strength Test Results

Type-2 mixture quantity, unit volume weight and compressive strength test results are shown in Table 23. The amounts of lime were taken as 0%, 20%, 40%, 60%, 80% and 100% of the pumice powder amount.

As can be seen in Table 23, in this section, the mixture with the highest strength was 2-3 type mixture and its compressive strength was calculated as 0.5 MPa.

#### 3.1.3. Type-3 Compressive Strength Test Results

Type-3 mixture quantity, unit volume weight and compressive strength test results are shown in Table 24. The amounts of lime was taken as 0%, 20%, 40%, 60%, 80% and 100% of the cement amount.

As shown in Table 24, the 3-1 mixture was the reference mortar of the binder paste. Only cement was used as binder in the reference mortar. In this section, the mixture with the highest strength regarding reference mortar was 3-2 and its compressive strength was found to be 20.1 MPa.

#### 3.1.4. Type-4 Compressive Strength Test Results

Compressive strength tests of the nine different types of mixture obtained from Type-4 mixtures were performed and the strengths of binder pastes were calculated.

##### Type-1 Compressive Strength Test Results

Type-4-1 mixture quantity, weight per unit of volume and compressive strength test results are shown in Table 25. In this section, the mixing ratio with the highest compressive strength in Type-1 (Table 22) mixture was considered.

Accordingly, in the mixing ratio with the highest compressive strength, the amount of lime were taken as 0%, 20%, 40%, 60%, 80% and 100% of the cement amount. As shown in Table 25, the highest strength in this section was a 4-1-1 type mixture with a compressive strength of 17.3 MPa.

##### Type-4-2 Compression Test Results

Type-4-2 mixture quantity, weight per unit of volume and compression test results are shown in Table 26. In this section, the mixing ratio having the highest compressive strength in the first type of mixture (Table 22) was considered.

In the mixing ratio, the compressive strength of which was found to be the highest, the amounts of lime were taken as 0%, 20%, 40%, 60%, 80% and 100% of the pumice powder amount. As shown in Table 26, the highest strength in this section was a 4-2-6 type mixture with a compressive strength of 20.1 MPa.

##### Type-4-3 Compression Test Results

Type-4-3 mixture quantity, weight per unit of volume and compression test results are shown in Table 27. In this section, the mixing ratio having the highest compressive strength in the first type of mixture (Table 22) was considered.

In the mixing ratio with the highest compressive strength, the amounts of lime were taken as 0%, 20%, 40%, 60%, 80% and 100% of the (cement + pumice powder) amount, respectively. As shown in Table 27, the highest strength in this section was a 4-3-1 type mixture with a compressive strength of 17.3 MPa.

##### Type-4-4 Compression Test Results

Type-4-4 mixture quantity, weight per unit of volume and compression test results are shown in Table 28. In this section, the mixing ratio having the highest compressive strength in the second type of mixture (Table 23) was considered.

In the mixing ratio, the compressive strength of which was found to be the highest, the amounts of cement were taken as 0%, 20%, 40%, 60%, 80% and 100% of the pumice powder amount. As shown in Table 28, the highest strength in this section was a 4-4-6 type mixture with a compressive strength of 20.2 MPa.

##### Type-4-5 Compression Test Results

Type-4-5 mixture quantity, weight per unit of volume and compression test results are shown in Table 29. In this section, the mixing ratio having the highest compressive strength in the second type of mixture (Table 23) was considered.

In the mixing ratio, the compressive strength of which was found to be the highest, the amounts of cement were taken as 0%, 20%, 40%, 60%, 80% and 100% of the lime amount. As seen in Table 29, the highest strength in this section was a 4-5-6 type mixture, the compressive strength of which was 7.6 MPa.

##### Type-4-6 Compression Test Results

Type-4-6 mixture quantity, weight per unit of volume and compression test results are shown in Table 30. In this section, the mixing ratio with the highest compressive strength in the second type of mixture (Table 23) was considered.

In the mixing ratio, the compressive strength of which was found to be the highest, the amounts of cement were taken as 0%, 20%, 40%, 60%, 80% and 100% of the (pumice powder + lime) amount. As seen in Table 30, the highest strength in this section was 4-6-6 type mixture, the compressive strength of which was 21.9 MPa.

##### Type-4-7 Compression Test Results

Type-4-7 mixture quantity, weight per unit of volume and compression test results are shown in Table 31. In this section, the mixing ratio, having the highest compressive strength in the third type of mixture (Table 24) was considered.

In the mixing ratio, the compressive strength of which was found to be the highest, the amounts of pumice powder were taken as 0%, 20%, 40%, 60%, 80% and 100% of the cement amount. As shown in Table 31, the highest strength in this section was a 4-7-1 type mixture with a compressive strength of 20.1 MPa.

##### Type-4-8 Compression Test Results

Type-4-8 mixture quantity, weight per unit of volume and compression test results are shown in Table 32. In this section, the mixing ratio having the highest compressive strength in the third type mixture (Table 24) was considered.

In the mixing ratio, the compressive strength of which was found to be the highest, the amounts of pumice powder were taken as 0%, 20%, 40%, 60%, 80% and 100% of the lime amount. As shown in Table 32, the highest strength in this section was a 4-8-1 type mixture with a compressive strength of 20.1 MPa.

##### Type-4-9 Compression Test Results

Type-4-9 mixture quantity, weight per unit of volume and compression test results are shown in Table 33. In this section, the mixing ratio having the highest compressive strength in the third type mixture (Table 24) was considered.

In the mixing ratio, the compressive strength of which was found to be the highest, the amounts of pumice powder were taken as 0%, 20%, 40%, 60%, 80% and 100% of the (cement + lime) amount. As shown in Table 33, the highest strength in this section is a 4-9-1 type mixture with a compressive strength of 20.1 MPa. The maximum compressive strengths of each type of mixture are shown in Table 34, taking into account all mixing ratios.

As can be seen in Table 34, regarding the fixation of optimum binder ratio, the highest strength was a 4-6-6 type mixture, the compressive strength of which was 21.9 MPa.

The compressive strength test results of all mixture types are shown in Figure 11.

As shown in Figure 11, regarding the fixation of optimum binder ratio, the highest strength was a 4-6-6 type mixture, the compressive strength of which was 21.9 MPa.

### 3.2. Sieve Analysis Results

Reference concrete (RC) and optimum binder concrete (PPCC) sieve analysis is shown in Table 35.

The reference concrete (RC) and optimum binder concrete (PPCC) sieve analysis graph is shown in Figure 12.

As shown in Figure 12, aggregate granulometry of concretes conformed to TS802 (2016) standard. Standard TS 802 (2016) (ACI 211.1-91) emphasizes that the gradation curve for such aggregates must lie between lines A16 and B16 or between lines B16 and C16.

### 3.3. Concrete Compressive and Bending Strength Test Results

Table 36 shows the compressive and bending strength test results of the reference concrete (RC) and the optimum binder concrete (PPCC). The results were presented for 7-day and 28-day periods. Three samples for each mixture design under each curing condition were taken, and the average of these three values was calculated.

The average compressive strength test results of reference concrete (RC) and optimum binder concrete (PPCC) are shown in Figure 13.

The average bending strength test results of reference concrete (RC) and optimum binder concrete (PPCC) are shown in Figure 14.

## 4. Conclusions

Pumice powder is a waste material that contributes to environmental pollution and waste landfills. The presence of pores in coarse pumice gives the extremely low compressive and bending strengths, and therefore, such materials cannot be used as aggregates in concrete production. On this basis, coarse pumice has limited applicability in the construction sector. In this study, the usability of pumice powder and lime in concrete production as a binding additive for concrete road pavement was investigated. A total of 72 types of concrete samples were composed with different mixing ratios, which were formed with cement, pumice powder and lime mixtures. The most appropriate ratios of cement, pumice powder and lime as the binding additive were determined in consequence of all the experiments. Following the determination of the optimum binder ratio, these new binder ratios were used in crushed limestone concrete production. Compressive and bending strength tests of the new concrete produced were performed. The concrete thus formed was named concrete containing cement, pumice powder and lime (PPCC). The normally produced concrete, without pumice powder and lime binder was selected as reference concrete (RC). The reference concrete and PPCC concrete were cured with standard water curing of 7 and 28 days. The following results were obtained in the study:As a result of the study, regarding the total binder amount of the most appropriate binder ratio, 50% was found to be cement, 30% pumice powder and 20% lime;In consequence of the study, the 20 ± 2 °C and 7–28 days average compressive strengths of reference concrete were found to be 33.8 MPa and 38.2 MPa, and the average bending strengths 4.2 MPa and 4.7 MPa;The 20 ± 2 °C and 7–28 days daily average compressive strengths of the concrete formed by optimum binding ratio were found to be 25.1 MPa and 28.3 MPa and the average bending strengths 3.2 MPa and 3.5 MPa. The results of the study showed the usability of PPCC in rigid pavement;Subsequent studies may conduct fatigue and impact tests on the samples produced in this work.

## Figures and Tables

**Figure 1 materials-12-02743-f001:**
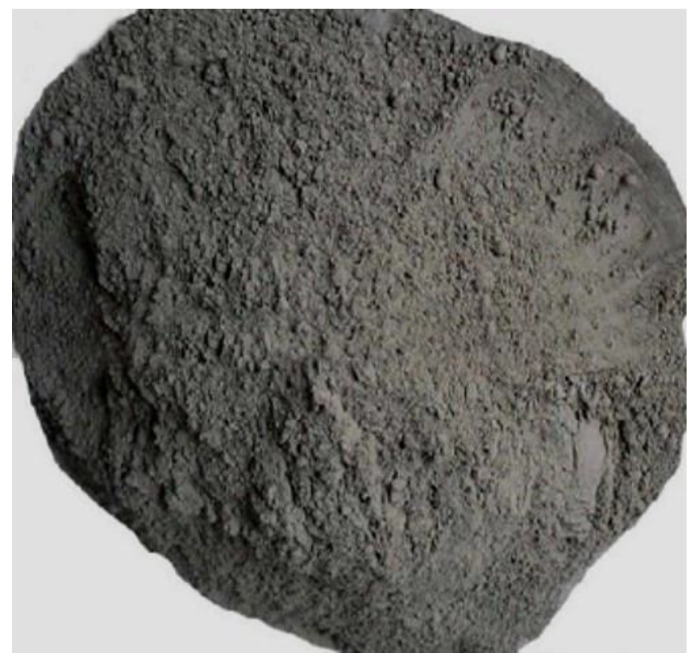
CEM I 42.5 R cement appearance.

**Figure 2 materials-12-02743-f002:**
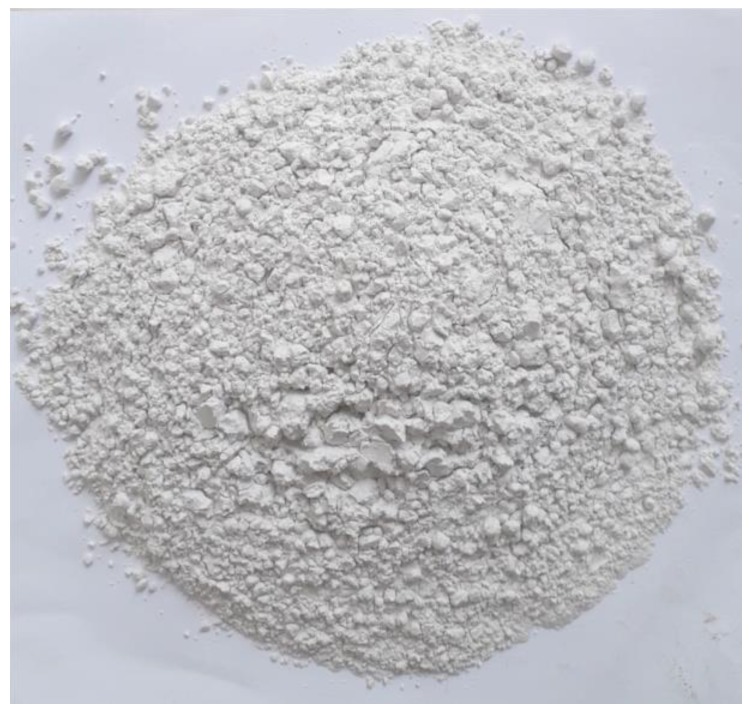
Pumice powder.

**Figure 3 materials-12-02743-f003:**
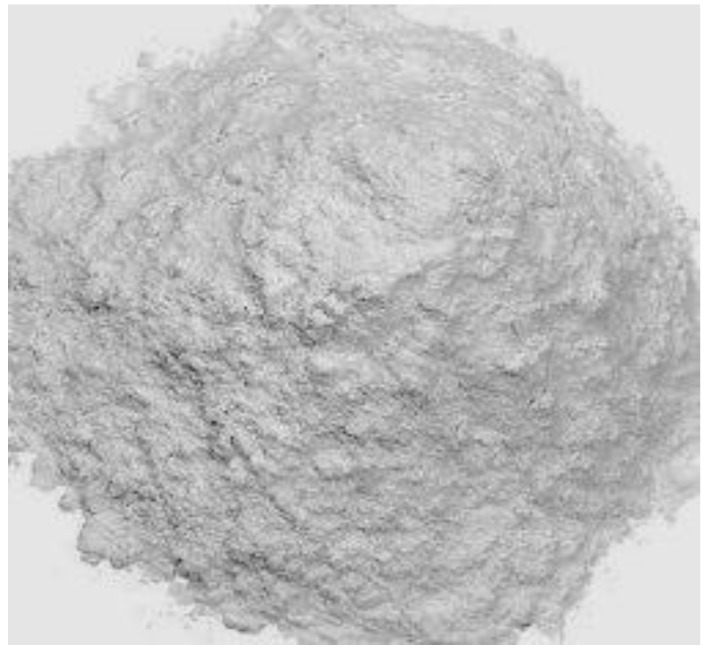
Lime.

**Figure 4 materials-12-02743-f004:**
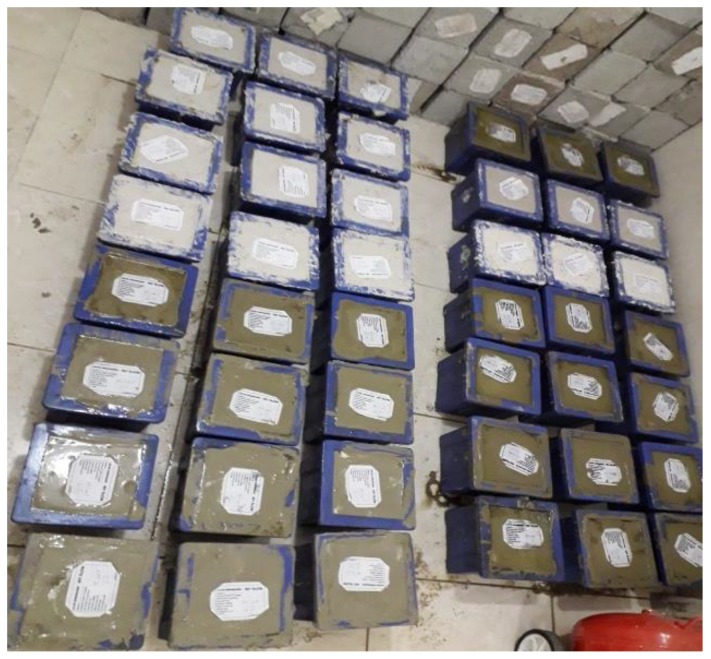
Mixtures prepared for optimum binding.

**Figure 5 materials-12-02743-f005:**
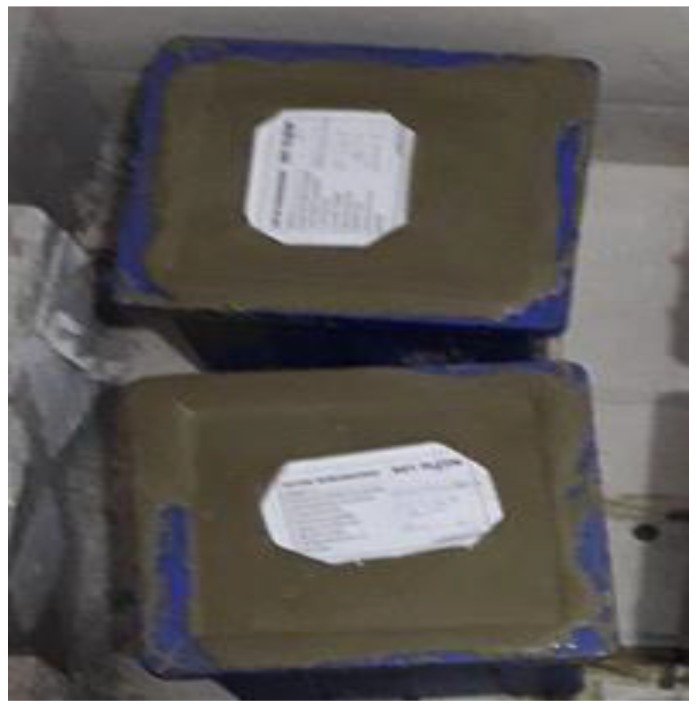
Binder reference mortar for optimum binder fixation.

**Figure 6 materials-12-02743-f006:**
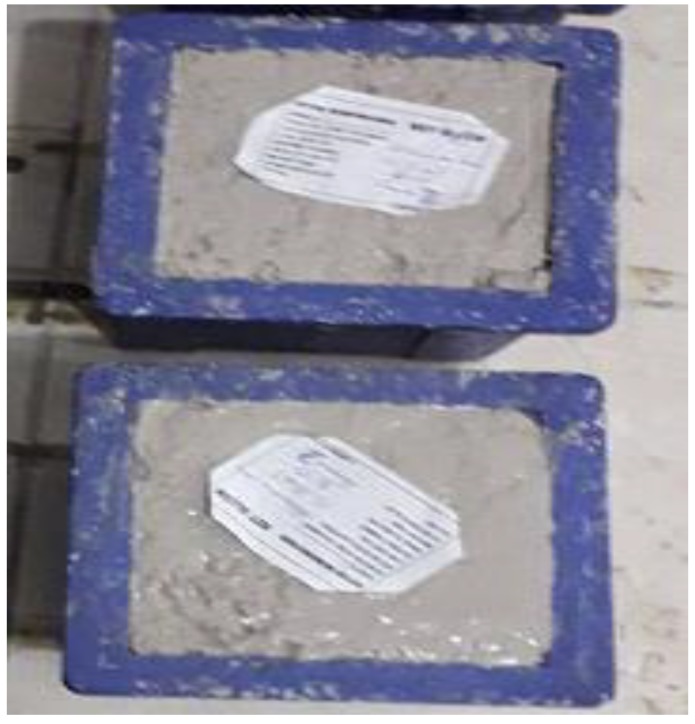
Pumice binder mortar for optimum binder fixation.

**Figure 7 materials-12-02743-f007:**
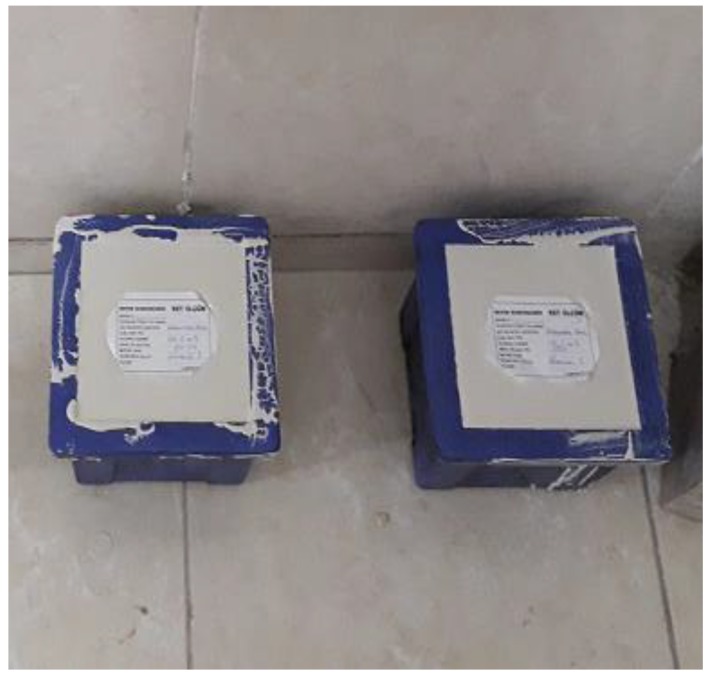
Lime binder mortar for optimum binder fixation.

**Figure 8 materials-12-02743-f008:**
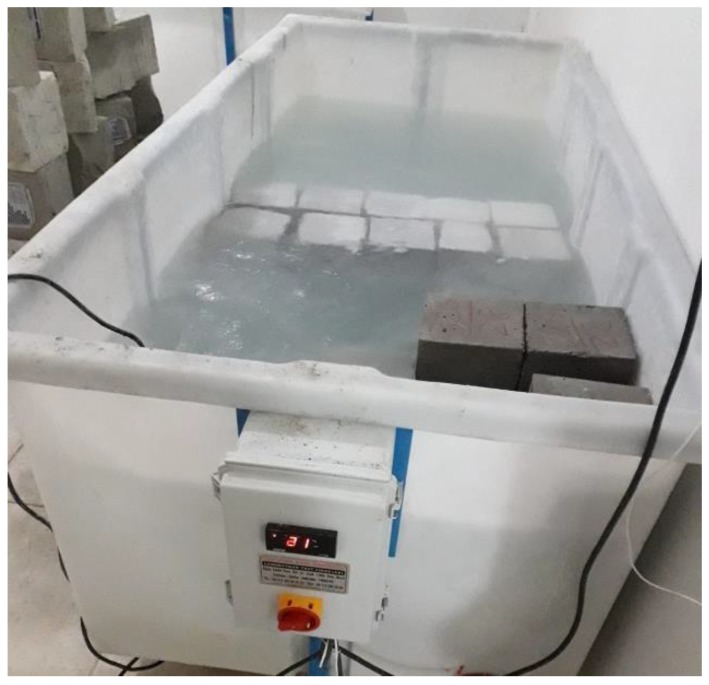
Curing pool.

**Figure 9 materials-12-02743-f009:**
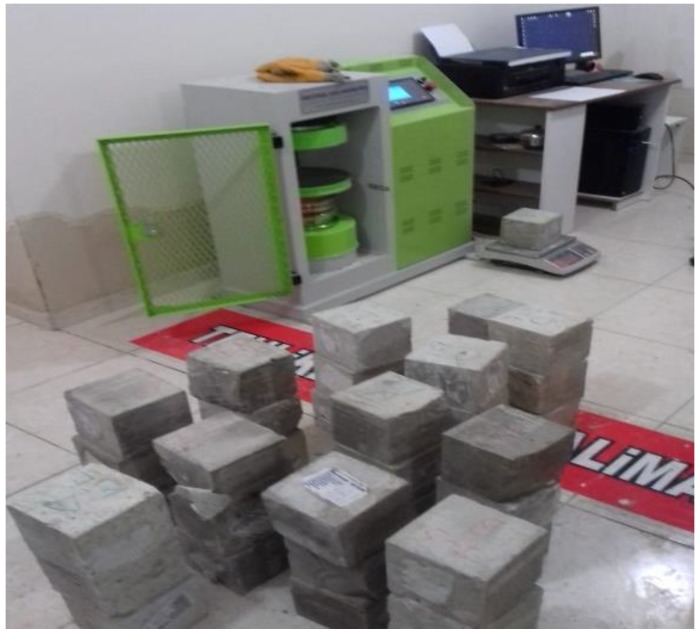
Samples prepared for compressive strength tests.

**Figure 10 materials-12-02743-f010:**
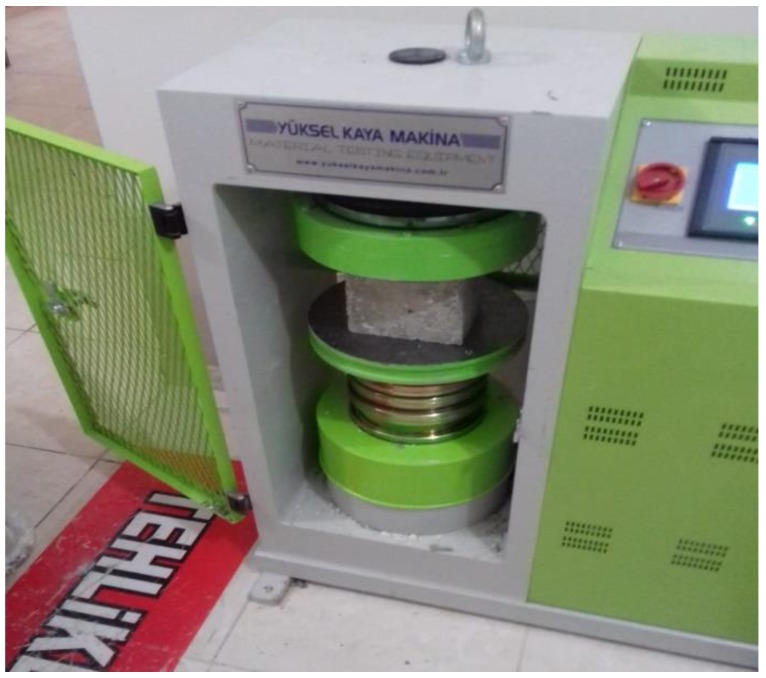
Appearance of samples in compressive strength tester.

**Figure 11 materials-12-02743-f011:**
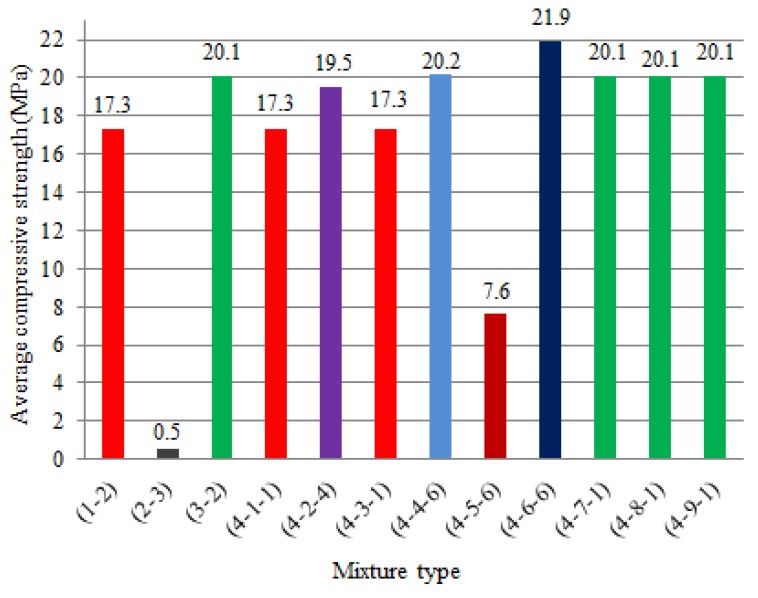
The average compressive strength test results of all mixture type.

**Figure 12 materials-12-02743-f012:**
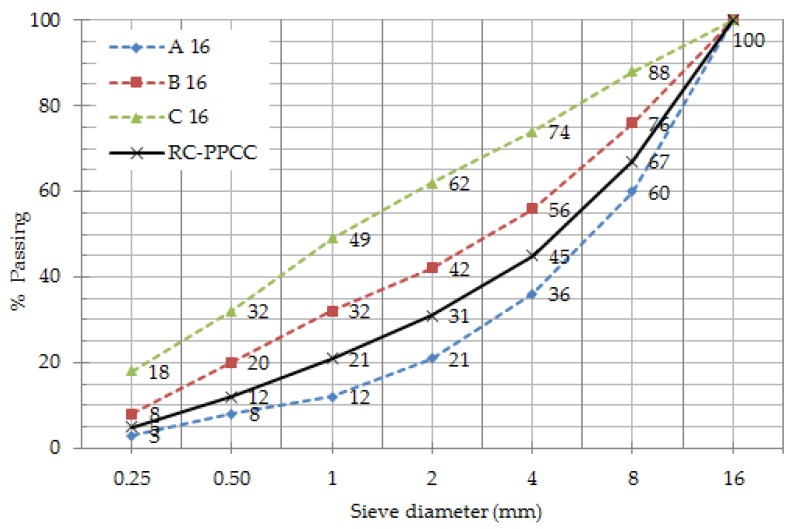
Reference concrete (RC) and optimum binder concrete (PPCC) sieve analysis chart.

**Figure 13 materials-12-02743-f013:**
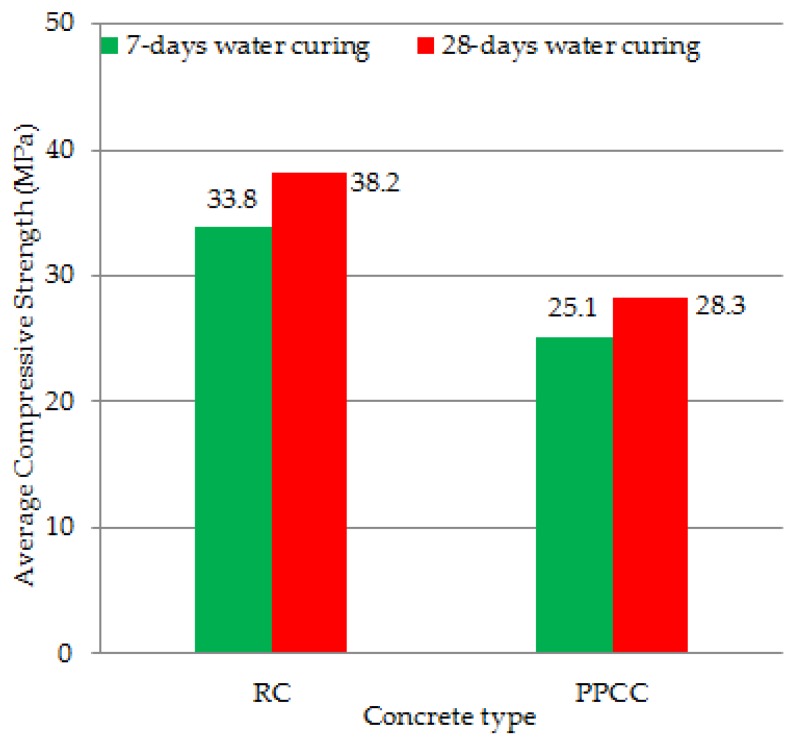
The average compressive strength results of RC and PPCC concrete.

**Figure 14 materials-12-02743-f014:**
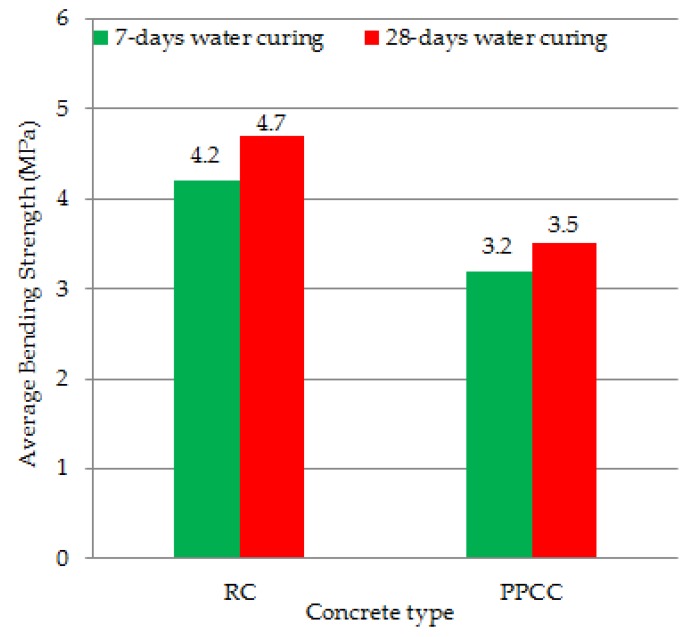
The average bending strength results of RC and PPCC concrete.

**Table 1 materials-12-02743-t001:** CEM I 42.5 R cement chemical properties.

Chemical Properties %	
SiO_2_	18.90
Al_2_O_3_	5.15
Fe_2_O_3_	3.36
CaO	63.59
MgO	1.57
SO_3_	2.65
Loss of Ignition	3.59
K_2_O	0.77
Na_2_O	0.40
Cl	0.02

**Table 2 materials-12-02743-t002:** Chemical properties of pumice and lime.

Sample	Loss of Ignition (%)	MgO (%)	Al_2_O_3_ (%)	SiO_2_ (%)	Na_2_O + K_2_O (%)	CaO (%)	Fe_2_O_3_ (%)	Ca (OH)_2_ (%)
Pumice	3	0.6	14	70	9	0.9	2.5	-
Lime	7	3	-	-	-	85	-	80

**Table 3 materials-12-02743-t003:** Physical and mechanical properties of pumice, cement, and lime.

Property	Pumice	Cement	Lime
Specific gravity (g/cm^3^)	2.415	3.130	1.800
Water absorption by weight (%)	22.59	-	-
Los Angeles abrasion (%)	73	-	-

**Table 4 materials-12-02743-t004:** Type-1 Mixing Ratios.

Mixture Type	Cement (C) (%)	Pumice Powder (%)	Lime (%)
1-1	100	0	0
1-2	80	20	0
1-3	60	40	0
1-4	40	60	0
1-5	20	80	0
1-6	0	100	0

**Table 5 materials-12-02743-t005:** Type-2 mixing ratios.

Mixture Type	Cement (C) (%)	Pumice Powder (%)	Lime (%)
2-1	0	100	0
2-2	0	80	20
2-3	0	60	40
2-4	0	40	60
2-5	0	20	80
2-6	0	0	100

**Table 6 materials-12-02743-t006:** Type-3 mixing ratios.

Mixture Type	Cement (C) (%)	Pumice Powder (%)	Lime (%)
3-1	100	0	0
3-2	80	0	20
3-3	60	0	40
3-4	40	0	60
3-5	20	0	80
3-6	0	0	100

**Table 7 materials-12-02743-t007:** Type-4-1 mixing ratios.

Mixture Type	Cement (C) (%)	Pumice Powder (%)	Lime (%)
4-1-1	80	20	0
4-1-2	64	20	16
4-1-3	48	20	32
4-1-4	32	20	48
4-1-5	16	20	64
4-1-6	0	20	80

**Table 8 materials-12-02743-t008:** Type-4-2 mixing ratios.

Mixture Type	Cement (C) (%)	Pumice Powder (%)	Lime (%)
4-2-1	80	20	0
4-2-2	80	16	4
4-2-3	80	12	8
4-2-4	80	8	12
4-2-5	80	4	16
4-2-6	80	0	20

**Table 9 materials-12-02743-t009:** Type 4-3 mixing ratios.

Mixture Type	Cement (C) (%)	Pumice Powder (%)	Lime (%)
4-3-1	80	20	0
4-3-2	66.6	16.7	16.7
4-3-3	57.1	14.3	28.6
4-3-4	50.0	12.5	37.5
4-3-5	44.4	11.2	44.4
4-3-6	40.0	10.0	50

**Table 10 materials-12-02743-t010:** Type 4-4 mixing ratios.

Mixture Type	Cement (C) (%)	Pumice Powder (%)	Lime (%)
4-4-1	0	60	40
4-4-2	10	50	40
4-4-3	17	43	40
4-4-4	22.5	37.5	40
4-4-5	26.7	33.3	40
4-4-6	30	30	40

**Table 11 materials-12-02743-t011:** Type 4-5 Mixing Ratios.

Mixture Type	Cement (C) (%)	Pumice Powder (%)	Lime (%)
4-5-1	0	60	40
4-5-2	6.7	60	33.3
4-5-3	11.4	60	28.6
4-5-4	15	60	25
4-5-5	17.8	60	22.2
4-5-6	20	60	20

**Table 12 materials-12-02743-t012:** Type 4-6 mixing ratios.

Mixture Type	Cement (C) (%)	Pumice Powder (%)	Lime (%)
4-6-1	0	60	40
4-6-2	16.7	50	33.3
4-6-3	28.6	42.8	28.6
4-6-4	37.5	37.5	25
4-6-5	44.4	33.4	22.2
4-6-6	50	30	20

**Table 13 materials-12-02743-t013:** Type 4-7 mixing ratios.

Mixture Type	Cement (C) (%)	Pumice Powder (%)	Lime (%)
4-7-1	80	0	20
4-7-2	64	16	20
4-7-3	48	32	20
4-7-4	32	48	20
4-7-5	16	64	20
4-7-6	0	80	20
4-8-1	80	0	20

**Table 14 materials-12-02743-t014:** Type 4-8 mixing ratios.

Mixture Type	Cement (C) (%)	Pumice Powder (%)	Lime (%)
4-8-1	80	0	20
4-8-2	80	4	16
4-8-3	80	8	12
4-8-4	80	12	8
4-8-5	80	16	4
4-8-6	80	20	0

**Table 15 materials-12-02743-t015:** Type 4-9 mixing ratios.

Mixture Type	Cement (C) (%)	Pumice Powder (%)	Lime (%)
4-9-1	80	0	20
4-9-2	66.6	16.7	16.7
4-9-3	57.1	28.6	14.3
4-9-4	50.0	37.5	12.5
4-9-5	44.4	44.4	11.2
4-9-6	40.0	50.0	10.0

**Table 16 materials-12-02743-t016:** Reference concrete material quantities.

Materials	Quantity (kg/m^3^)
Portland Cement	450
Crushed Stone (0–4 mm)	792
Crushed Stone (4–8 mm)	388
Crushed Stone (8–16 mm)	581
Water	189
Total	2400

**Table 17 materials-12-02743-t017:** Amounts of concrete material at optimum binder ratio.

Materials	Quantity (kg/m^3^)
Portland Cement	225
Pumice powder (0–0.04 mm)	135
Lime	90
Crushed Stone (0–4 mm)	792
Crushed Stone (4–8 mm)	388
Crushed Stone (8–16 mm)	581
Water	189
Total	2400

**Table 18 materials-12-02743-t018:** Coating concrete slump values.

Type of Concrete	Slump, cm	
	Minimum	Maximum
Coating, Precast, Roller Compacted Concrete	1–2	3–4
Mass Concrete	2.5	5.0
Reinforced Concrete (Very Equipped)	2.5	10
Reinforced Concrete (Less Equipped)	2.5	7.5

**Table 19 materials-12-02743-t019:** Approximate water/cement ratios according to concrete compressive strengths.

Compressive Strength (28 Days) 150 × 300 mm Cylinder (MPa)	Water/Cement Ratio	
	Non-Air Entrained Concrete	Air Entrained Concrete
15	0.79	0.70
20	0.69	0.60
25	0.61	0.52
30	0.54	0.45
35	0.47	0.39
40	0.42	0.33
45	0.37	0.29
50	0.33	0.25

**Table 20 materials-12-02743-t020:** Average compressive strengths according to concrete classes.

Concrete Category	Characteristic Compressive Strength (MPa)		Target Compressive Strength (MPa)	
	Characteristic Cylinder (150 × 300) mm Compressive Strength	Equivalent Cube (150 × 150 × 150) mm Compressive Strength	(150 × 300) mm Cylinder	(150 × 150 × 150) mm Cube
C30/37	30	37	36	43

**Table 21 materials-12-02743-t021:** Mass of test specimens for normal aggregates.

Aggregate Grain Size (D) (up to) mm	Sample Amount Required for Experiment (minimum) kg
63	40
32	10
16	2.6
8	0.6
<4	0.2

**Table 22 materials-12-02743-t022:** Type-1 mixture quantity, unit volume weight and compressive strength test results.

Mixture Type	BHA (t/m^3^)	Cement (C) (kg)	Pumice Powder (kg)	Lime (kg)	Water (kg)	Average Compressive Strength (MPa)
1-1	5.834	1500	0	0	900	**29.7**
1-2	5.534	1200	300	0	900	17.3
1-3	5.503	900	600	0	900	12.7
1-4	5.292	600	900	0	900	10.1
1-5	5.058	300	1200	0	900	7.6
1-6	4.850	0	1500	0	900	0.2

**Table 23 materials-12-02743-t023:** Type-2 mixture quantity, unit volume weight and compressive strength test results.

Mixture Type	BHA (t/m^3^)	Cement (C) (kg)	Pumice Powder (kg)	Lime (kg)	Water (kg)	Average Compressive Strength (MPa)
2-1	4.850	0	1500	0	900	0.2
2-2	4.863	0	1200	300	900	0.3
2-3	4.874	0	900	600	900	**0.5**
2-4	4.916	0	600	900	900	0.4
2-5	4.945	0	300	1200	900	0.3
2-6	4.975	0	0	1500	900	0.2

**Table 24 materials-12-02743-t024:** Type-3 mixture quantity, unit volume weight and compressive strength test results.

Mixture Type	BHA (t/m^3^)	Cement (C) (kg)	Pumice Powder (kg)	Lime (kg)	Water (kg)	Average Compressive Strength (MPa)
3-1	5.834	1500	0	0	900	**29.7**
3-2	5.806	1200	0	300	900	20.1
3-3	5.663	900	0	600	900	14.5
3-4	5.550	600	0	900	900	13.0
3-5	5.325	300	0	1200	900	8.1
3-6	4.975	0	0	1500	900	0.2

**Table 25 materials-12-02743-t025:** Type-1 mixture quantity, weight per unit of volume and compression test results.

Mixture Type	BHA (t/m^3^)	Cement (C) (kg)	Pumice Powder (kg)	Lime (kg)	Water (kg)	Average Compressive Strength (MPa)
4-1-1	5.534	1200	300	0	900	**17.3**
4-1-2	5.666	960	300	240	900	15.6
4-1-3	5.626	720	300	480	900	9.6
4-1-4	5.420	480	300	720	900	4.3
4-1-5	5.198	240	300	960	900	1.2
4-1-6	4.945	0	300	1200	900	1.1

**Table 26 materials-12-02743-t026:** Type-2 mixture quantity, weight per unit of volume and compression test results.

Mixture Type	BHA (t/m^3^)	Cement (C) (kg)	Pumice Powder (kg)	Lime (kg)	Water (kg)	Average Compressive Strength (MPa)
4-2-1	5.534	1200	300	0	900	17.3
4-2-2	5.559	1200	240	60	900	15.4
4-2-3	5.624	1200	180	120	900	17.4
4-2-4	5.717	1200	120	180	900	19.5
4-2-5	5.780	1200	60	240	900	18.0
4-2-6	5.806	1200	0	300	900	**20.1**

**Table 27 materials-12-02743-t027:** Type-4-3 mixture quantity, weight per unit of volume and compression test results.

Mixture Type	BHA (t/m^3^)	Cement (C) (kg)	Pumice Powder (kg)	Lime (kg)	Water (kg)	Average Compressive Strength (MPa)
4-3-1	5.534	1200	300	0	900	**17.3**
4-3-2	5.516	999	250.5	250.5	900	15.5
4-3-3	5.508	856.5	214.5	429	900	9.2
4-3-4	5.385	750	187.5	562.5	900	4.2
4-3-5	5.303	666	168	666	900	1.2
4-3-6	5.225	600	150	750	900	1.1

**Table 28 materials-12-02743-t028:** Type-4-4 mixture quantity, weight per unit of volume and compression test results.

Mixture Type	BHA (t/m^3^)	Cement (C) (kg)	Pumice Powder (kg)	Lime (kg)	Water (kg)	Average Compressive Strength (MPa)
4-4-1	4.874	0	900	600	900	0.5
4-4-2	5.152	150	750	600	900	1.3
4-4-3	5.166	255	645	600	900	3.5
4-4-4	5.492	337.5	562.5	600	900	7.0
4-4-5	5.607	400.5	499.5	600	900	15.6
4-4-6	5.710	450	450	600	900	**20.2**

**Table 29 materials-12-02743-t029:** Type-4-5 mixture quantity, weight per unit of volume and compression test results.

Mixture Type	BHA (t/m^3^)	Cement (C) (kg)	Pumice Powder (kg)	Lime (kg)	Water (kg)	Average Compressive Strength (MPa)
4-5-1	4.874	0	900	600	900	0.5
4-5-2	4.891	100.5	900	499.5	900	1.5
4-5-3	4.915	171	900	429	900	1.9
4-5-4	4.957	225	900	375	900	3.1
4-5-5	5.013	267	900	333	900	5.2
4-5-6	5.058	300	900	300	900	**7.6**

**Table 30 materials-12-02743-t030:** Type-4-6 mixture quantity, weight per unit of volume and compression test results.

Mixture Type	BHA (t/m^3^)	Cement (C) (kg)	Pumice Powder (kg)	Lime (kg)	Water (kg)	Average Compressive Strength (MPa)
4-6-1	4.874	0	900	600	900	0.5
4-6-2	4.913	250.5	750	499.5	900	0.9
4-6-3	4.951	429	642	429	900	4.9
4-6-4	4.997	562.5	562.5	375	900	10.2
4-6-5	5.015	666	501	333	900	15.6
4-6-6	5.037	750	450	300	900	**21.9**

**Table 31 materials-12-02743-t031:** Type-4-7 mixture quantity, weight per unit of volume and compression test results.

Mixture Type	BHA (t/m^3^)	Cement (C) (kg)	Pumice Powder (kg)	Lime (kg)	Water (kg)	Average Compressive Strength (MPa)
4-7-1	5.806	1200	0	300	900	**20.1**
4-7-2	5.607	960	240	300	900	15.6
4-7-3	5.492	720	480	300	900	7.0
4-7-4	5.166	480	720	300	900	3.5
4-7-5	5.152	240	960	300	900	1.3
4-7-6	4.863	0	1200	300	900	0.3

**Table 32 materials-12-02743-t032:** Type-4-8 mixture quantity, weight per unit of volume and compression test results.

Mixture Type	BHA (t/m^3^)	Cement (C) (kg)	Pumice Powder (kg)	Lime (kg)	Water (kg)	Average Compressive Strength (MPa)
4-8-1	5.806	1200	0	300	900	**20.1**
4-8-2	5.780	1200	60	240	900	18.0
4-8-3	5.717	1200	120	180	900	19.5
4-8-4	5.624	1200	180	120	900	17.4
4-8-5	5.559	1200	240	60	900	15.4
4-8-6	5.534	1200	300	0	900	17.3

**Table 33 materials-12-02743-t033:** Type-4-9 mixture quantity, weight per unit of volume and compression test results.

Mixture Type	BHA (t/m^3^)	Cement (C) (kg)	Pumice Powder (kg)	Lime (kg)	Water (kg)	Average Compressive Strength (MPa)
4-9-1	5.806	1200	0	300	900	**20.1**
4-9-2	5.516	999	250.5	250.5	900	15.5
4-9-3	5.479	856.5	429	214.5	900	9.3
4-9-4	5.316	750	562.5	187.5	900	3.9
4-9-5	4.991	666	666	168	900	1.0
4-9-6	4.976	600	750	150	900	0.7

**Table 34 materials-12-02743-t034:** Maximum compressive strength results for optimum binder fixation.

Mixture Type	BHA (t/m^3^)	Cement (C) (kg)	Pumice Powder (kg)	Lime (kg)	Water (kg)	Average Compressive Strength (MPa)
1-2	5.534	1200	300	0	900	17.3
2-3	4.874	0	900	600	900	0.5
3-2	5.806	1200	0	300	900	20.1
4-1-1	5.534	1200	300	0	900	17.3
4-2-4	5.717	1200	120	180	900	19.5
4-3-1	5.534	1200	300	0	900	17.3
4-4-6	5.710	450	450	600	900	20.2
4-5-6	5.058	300	900	300	900	7.6
4-6-6	5.037	750	450	300	900	**21.9**
4-7-1	5.806	1200	0	300	900	20.1
4-8-1	5.806	1200	0	300	900	20.1
4-9-1	5.806	1200	0	300	900	20.1

**Table 35 materials-12-02743-t035:** Reference concrete (RC) and optimum binder concrete (PPCC) sieve analysis.

Sieve Diameter (mm)	On Sieve Remaining Weight (gr)	On Sieve Total Remaining Weight (gr)	On Sieve Total Remaining Weight (%)	Remaining Under Sieve (%)
16	-	-	-	100
8	990	990	33	67
4	660	1650	55	45
2	420	2070	69	31
1	300	2370	79	21
0.5	270	2640	88	12
0.25	210	2850	95	5

**Table 36 materials-12-02743-t036:** The average compressive and bending strength test results of concrete types.

Concrete Type	BHA (t/m^3^)	Curing Type	Average Compressive Strength (MPa)	Average Bending Strength (MPa)
RC	2.290	7 days 20 ± 2 °C water curing	33.8	4.2
		28 days 20 ± 2 °C water curing	38.2	4.7
PPCC	2.150	7 days 20 ± 2 °C water curing	25.1	3.2
		28 days 20 ± 2 °C water curing	28.3	3.5

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
