# Peer review of "The Usability of Pumice Powder as a Binding Additive in the Aspect of Selected Mechanical Parameters for Concrete Road Pavement"

_materials, 2019, doi:10.3390/ma12172743_

Round 1
Reviewer 1 Report
The tests carried out are quite wide, however the final results should be grouped together in a single graph in order to see the deviation they have.
I think the intermediate results are irrelevant and do not provide any information.
On the other hand there are international regulations that should be referred to, since being an investigation to publish must be applicable as a whole.
It would be good to perform an aging test of the material to see what is the durability, this being a very simple test to make and provides important information.
There are some typographical errors that would be recommended to correct as in the 32 line m3 for example
Reviewer 2 Report
Dear authors,
the paper presents a lot of experimental work which was provided. The idea to substitute cement by alternative additives like pumice powder follows actual trends either with respect to reduce waste production or to produce low-emission cements. The topic is therefore actual.
The weaknesses of the paper are:
necessity to have a proof reading providing some modifications in some sentences to be better readable some changes in structuring the text to avoid repeating sentences several times clarify that substitution of cement by pumice powder does not affect the concrete consistency and the same w/c can be used for all variants. From my experience with fly-ashes or slags this normally is not true and if cement is in larger amounts replaced, it needs normally some fine-tuning of w/c. Or at least it shall be mentioned that consistency and workability is not influenced by the substitution.
I am not surprised that the strength properties dropped don since pumice particles are known as potentially interesting material for light-weight concrete. Usually this is related to lower strength properties as well. Maybe some information about chemical composition of pumice powder (and/or the mix of this powder with cement and lime) and potential impact on the concrete performance would be helpful as well.
Finally in the conclusions the findings are summarized once again, but it is not clear if pumice powder has some potential to be practically used, if you are recommending it or you are proposing different actions - e.g. will some chemistry like plastisizers have better impact on resulting concrete properties? The conclusions shall be somehow improved.
My comments are in the attached original file.

Reviewer 3 Report
The report contains original research but requires additions / corrections:
The requirements for concrete intended for airport pavements are more stringent than for structural concrete. Designing a common composition of the mixture, without taking into account differences (even such as loads) is superficial and erroneous. Please specify for which type of concrete the design assumptions were adopted. If airport pavements appear in the title of the article, please refer to this type of pavement consistently. Pumice and pumice powder were used interchangeably in the article, please standardize. Please explain in the article how lime and pumice powder were used in the mixtures (by weight, fraction or volume relative to the cement content). Please provide physical and mechanical parameters for cement and lime for comparison purposes (Table 3). Figures 5, 6 and 7 are of poor quality and are a kind of repetition of Figure 4. I think that they should be removed from the text because they are not relevant. Text below Figures 5, 6 and 7 ("As shown in table 4 .... in the reference mortar", "As shown in table 4 .... in the mixture mortar 1-6", "As shown in table 5 .... in the mixture mortar 2-6 ") is not needed. The text does not describe the procedure for testing compressive strength and bending strength. Please indicate for what period (7 or 28 days) the results were presented in points 3.1.1 - 3.1.4.9. Sections 3.1.1 - 3.1.4.9 lack an assessment of the significance of the obtained compressive strength results. What concrete compressive strength was given?For a single sample or is the average of 3 measurements? If a series of concrete has been evaluated on the basis of three samples, it is necessary to determine whether the results obtained are statistically significant. Figures 12 and 13 show the repetition of the results from Table 36, if they should already be extended by the significance assessment. The last sentence of the conclusions "The results of ... in rigid paving and building construction" is too far-reaching. The title of the article refers only to road pavements, so why does structural concrete also appear in the conclusions? In general, inference based on only two selected mechanical parameters is questionable.The article deals with extensive experimental topics related to only two selected mechanical parameters, therefore it is necessary to change the title of the article. Proposed title: "The usability of pumice powder as a binding additive in the aspect of selected mechanical parameters for Concrete Road Pavement".
Round 2
Reviewer 1 Report
All comments that have been submitted to the author have been made, except for the aging of the material.
Reviewer 2 Report
After first review now the manuscript is OK for me.
Reviewer 3 Report
The text of the paper still lacks an assessment of the results obtained.1. Presented results in points 3.1.1. - 3.1.4.9 still contain only "shear strength". It is not known if this is a result for a single sample or is the average of 3 measurements? If a series of concrete has been evaluated on the basis of three samples, it is necessary to determine whether the results obtained are statistically significant. Then write: "average compressive strength" and IMPERATIVE to add at least the "standard deviation" column.
2. In point 3.3. please indicate for what period the results are presented. Also, is this the result for a single sample or is the average of 3 measurements? If a series of concrete has been evaluated on the basis of three samples, it is necessary to determine whether the results obtained are statistically significant. Then write: "average compressive strength" or "average blending strength". IMPERATIVE, please add at least the "standard deviation" column - after all, the standards provide ranges of results scattering that are acceptable and, consequently, necessary to evaluate the results obtained.
